# Effect of UV Filters during the Application of Pulsed Light to Reduce *Lactobacillus brevis* Contamination and 3-Methylbut-2-ene-1-thiol Formation While Preserving the Physicochemical Attributes of Blonde Ale and Centennial Red Ale Beers

**DOI:** 10.3390/foods12040684

**Published:** 2023-02-04

**Authors:** Anubhav Pratap-Singh, Andrew Suwardi, Ronit Mandal, Joana Pico, Simone D. Castellarin, David D. Kitts, Anika Singh

**Affiliations:** 1Food Nutrition and Health Program, Faculty of Land & Food Systems, The University of British Columbia, Vancouver, BC V6T 1Z4, Canada; 2Wine Research Centre, Faculty of Land and Food Systems, University of British Columbia, 2205 East Mall, Vancouver, BC V6T 1Z4, Canada; 3Natural Health and Food Products Research Group, Centre for Applied Research and Innovation (CARI), British Columbia Institute of Technology, 4355 Mathissi Pl, Burnaby, BC V5G 4S8, Canada

**Keywords:** pulsed light, beverage decontamination, UV filters, beer processing, quality parameters, lightstruck, 3-MBT

## Abstract

Pulsed light (PL) is a novel, non-thermal technology being used to control the microbial spoilage of foods and beverages. Adverse sensory changes, commonly characterized as “lightstruck”, can occur in beers when exposed to the UV portion of PL due to the formation of 3-methylbut-2-ene-1-thiol (3-MBT) upon the photodegradation of iso-α-acids. This study is the first to investigate the effect of different portions of the PL spectrum on UV-sensitive beers (light-colored blonde ale and dark-colored centennial red ale) using clear and bronze-tinted UV filters. PL treatments with its entire spectrum, including the ultraviolet portion of the spectrum, resulted in up to 4.2 and 2.4 log reductions of *L. brevis* in the blonde ale and centennial red ale beers, respectively, but also resulted in the formation of 3-MBT and small but significant changes in physicochemical properties including color, bitterness, pH, and total soluble solids. The application of UV filters effectively maintained 3-MBT below the limit of quantification but significantly reduced microbial deactivation to 1.2 and 1.0 log reductions of *L. brevis* at 8.9 J/cm^2^ fluence with a clear filter. Further optimization of the filter wavelengths is considered necessary to fully apply PL for beer processing and possibly other light-sensitive foods and beverages.

## 1. Introduction

Beer is the third most popular beverage consumed globally after water and tea [1]. There is an upward trend in global beer production where an estimated 1.91 billion hectoliters were produced in 2019 [2]. Although current breweries have adopted better food safety and stricter sanitation programs, beer spoilage due to external sources of contamination remains a common problem for the beer industry. For example, the bacterial and fungal contamination of beer can lead to spoilage indexed by the occurrence of turbidity, acidification, and the production of undesirable aromatic compounds, such as diacetyl and hydrogen sulfide. Beer naturally has several intrinsic properties that can collectively provide an unfavorable environment for microorganism viability, including high CO_2_ content (~5 g/L), an ethanol content of 4.5–5.5% alcohol by volume (ABV), a low pH (3.8–4.7), and a low O_2_ content (<0.1 ppm) [3]. Additionally, the presence of iso-α-acids in beer (derived from the thermal degradation of hops’ compounds) at approximately 17–55 ppm has antimicrobial activity against Gram-positive bacteria, specifically lactic acid bacteria (LAB) such as *Lactobacillus brevis*, *Pediococcus damnosus, Lactobacillus casei, Lactobacillus curvatus*, and *Lactobacillus pantarum,* which are known as major beer spoilage microorganisms [4]. Craft beers made in microbreweries are particularly prone to beer spoilage bacteria as they tend to minimize pasteurization and microfiltration processes in order to showcase the pure flavor of beer [5]. With the popularity of craft beers on the rise in the past decade, an efficient decontamination process that prevents deterioration of beer organoleptic qualities is required [6].

Pulsed light (PL) is a novel and non-thermal food processing technology that is considered a potential alternative to pasteurization or microfiltration. PL kills microorganisms using short bursts of intense light applied on foods through photo-chemical, photo-thermal, and photo-physical effects [7]. PL achieves a high peak power by storing electricity in a capacitor for a relatively long time and releasing it in a shorter period with a wide spectrum consisting of UV rays (λ = 200–400 nm), visible light (λ = 400–700 nm), and IR rays (λ = 700–1100 nm). PL as a technology is well established in the literature for surface decontamination; for example, it can provide up to five log reductions of *L. monocytogenes* [8,9].

PL has successfully been applied to beverages, such as fruit juices, milk, tea, or wine [10,11,12,13]. However, notable challenges still exist for processing beverages with PL, since the application is limited by its penetration depth. Recent efforts in the PL processing of beverages have focused on optimizing the fluence (energy received by the product per unit area per unit time) to control some of the deteriorative light-induced reactions in beverages. Most efforts concentrate on finding optimal fluence levels that can perform microbial reduction while also producing minimal changes in product quality. Mandal et al. [12] reported 3.8 J/cm^2^ as the optimal fluence for processing black tea to minimize color change, phenolic degradation, and anti-oxidant activity loss. Mohammadi et al. [14] found a fluence of 2–4 J/cm^2^ optimal for the retention of >80% anthocyanins during the PL treatment of model wine solutions. Wiktor et al. [15] also reported 3.8 J/cm^2^ as the optimal fluence for processing model phenolic solutions. Hosseini et al. [16] observed more than three log reductions in *Escherichia coli* K12 when non-alcoholic beer was subjected to 10 J/cm^2^ pulsed laser light using 266 nm. However, no published research has investigated the application of PL on beer for routine decontamination purposes. For beverages, the application of PL on beer in a continuous production system poses an attractive solution to limit beer spoilage.

Despite the potential advantage of using PL to process beer, there are also complex challenges that could have an adverse effect on beer quality; hence, any modification of this application requires an evaluation of the equipment used to enable microbial reduction without changing quality parameters. First, beer has a low UV transmittance and short penetration depth, requiring specialized systems capable of handling liquids in thin profiles. Mandal and Pratap-Singh [17] developed continuous-flow pulsed UV light reactors for processing liquid foods in annular tube and coiled tube configurations and found that coiled tube reactors offered a higher efficiency compared to annular tube reactor designs. However, research has also established that beer exposure to UV and visible light (up to 500 nm) can trigger the degradation reactions of the iso-α-acids present in beer leading to the formation of 3-methylbut-2-ene-1-thiol (3-MBT), a phenomenon commonly known as “lightstruck” [18,19]. 3-MBT is an undesirable volatile compound with an aroma associated with skunk spray and has an extremely low sensory threshold of 2–7 ng/L [20,21]. Limiting the exposure to UV wavelengths is critical in the successful deployment of a PL design system on light-sensitive beverages such as beer. 

To limit the formation of 3-MBT by removing wavelengths below 400 nm, clear and bronze-tinted polycarbonate solid sheets (PSS) were tested as optical filters in this study. The overall objective of this research was to investigate the effect of filtering out the UV part of the wavelengths of PL on the microbial reduction in *L. brevis*, concentrations of 3-MBT, and changes in the physicochemical properties of beer, namely color, bitterness, pH, and total soluble solids (TSS). It was hypothesized that the PL treatment of beers with UV filters would limit 3-MBT concentrations compared to treatment without any filters. The use of a bronze-tinted filter was hypothesized to reduce 3-MBT more effectively than the clear filter, since the bronze-tinted filter not only removes UV akin to a clear filter but also inhibits visible light transmission between 400 and 500 nm, a known trigger wavelength range for 3-MBT formation, and reduces the transmission of other visible and infrared wavelengths. To the best of our knowledge, there is currently no study of pulsed light on beer.

## 2. Materials and Methods

### 2.1. Beer Samples

Two beer styles, blonde ale (BA, lot code 33462) and centennial red ale (CA, lot code 21447) were sourced from Faculty Brewing Co., a microbrewery located in Vancouver, BC, Canada. Both beer styles were received at the lab packaged in 473 mL cans. Upon receipt, the beers were stored at 4 °C until needed. Both beer styles are categorized as ales and top-fermenting beers but have different ingredients, recipes, and organoleptic qualities. The BA was visually lighter in color and more transparent (Appendix A) and contained 5.1% ABV and 20 international bitterness units (IBU). The CA was visually darker with a red-brown hue (Appendix A) and contained 4.8% ABV and 30 IBU. Appendix A outlines and compares the different starch sources, hops, and yeasts used in each beer style, published by the microbrewery. Both beer styles only used malts as starch sources with no adjuncts added. BA was hopped with magnum hops (USA), while CA was hopped with centennial hops (USA). The beers were fermented by different strains of pre-packaged *S. cerevisiae* strains (Lesaffre, France). The malts for both beer styles were similarly mashed at 66 °C for 6 min (Appendix A). Magnum hops were added to the BA wort only once and boiled for 60 min. For CA, centennial hops were added to the wort three times and boiled for 1, 15, and 60 min.

### 2.2. Pilot-Scale Pulsed Light Operation

A continuous thin-profile pilot-scale PL equipment (Solaris Disinfection Inc., Mississauga, ON, Canada) fitted with a coiled tube (CT) reactor, described earlier by Mandal and Pratap-Singh [17], was set up and used in the laboratory darkroom of the Food Process Engineering Lab at the University of British Columbia for PL treatment (Figure 1a). The PL lamp was 8 mm in diameter and 62 cm in length, supplying polychromatic light from 200 to 1100 nm wavelengths at 2.0 kV. The flash rate was maintained at 3 Hz. The distance between the lamp and the coil chamber was maintained at 5.9 cm, including the filter placed below the coil chamber. The CT reactor was made of quartz due to its high optical transmittivity for a broad range of wavelengths. The CT reactor was 56.3 cm in length, with a 0.95 mm wall thickness, 0.9 cm inner tube diameter, 1.19 cm outer tube diameter, 26.2 mm inner coil diameter, 48.5 mm outer coil diameter, 38 mm helical diameter, and 49 turns. The minimum sample volume required was 300 mL to simulate a controlled, continuous process. The flash rate, treatment time, and treatment initiation were controlled via a wired controller located outside the darkroom supplied by Solaris Disinfection Inc. A Masterflex L/S Economy Variable-Speed Drive (Masterflex, Quebec, QC, Canada) unidirectional peristaltic fluid pump was used to pump samples from the input tank through the coil chamber at different speeds. The intake tube was submerged into a slightly slanted input tank. A customized stainless-steel parabolic metal cover was fabricated (False Creek Fabrication, Vancouver, BC, Canada) and placed over the coil chamber. The cover’s interior wall was made shiny to reflect the pulsed light to the top of the coil chamber. The cover had holes on two sides to allow the input and output tubes through. Due to the tubing’s transparency outside of the coil chamber, they were all covered with aluminum foil to avoid uncontrolled light exposure as the fluid traveled.

In this study, 3 mm-thick transparent and bronze-tinted PSS (POLYSHINE, Shanghai Pincheng Plastics Co., Ltd., Shanghai, China) were used as clear and bronze-tinted filters, respectively. The solid sheets were cut to 30 inches in length and 6 inches in width, then placed above the PL frame. The chamber coil was rested on top of the filter. The clear filter allowed approximately 90% light transmission at 400–1100 nm wavelengths while limiting UV transmission at wavelengths below 400 nm (Figure 1b). The bronze-tinted filter allowed visible and NIR wavelength transmissions at lower levels than the clear filter while limiting UV wavelength transmissions similarly to that of the clear filter. Before treatment, the filters were sprayed with 70% ethanol solution and wiped with Kimwipes to remove dust and other debris. The fluence per flash received by the coil chamber with no filter, the clear filter, and the bronze-tinted filter were 18.61 ± 0.39, 9.92 ± 0.27, and 1.47 ± 0.12 mJ/cm^2^, respectively. The transmission through the filters, as well as the fluence through the reactors, was measured using a 1 cm^2^ aperture polyelectric head sensor (PE80BF-DIF-C, Ophir-Spiricon LLC, North Logan, UT, USA) with a Nova II display (OphirSpiricon LLC, North Logan, UT, USA) at the proximal end of the reactor, as described by Mandal and Pratap-Singh [17].

### 2.3. Treatment Types

Before each treatment, the PL fluid chamber and tubing system were sanitized with 3% hydrogen peroxide solution (Sigma, Toronto, ON, Canada) for 5 min and then rinsed again with sterile distilled water. In the dark, the operator turned on the pump and started the PL treatment. Once the treatment was over, the operator entered the darkroom and manually turned off the pump. 

Two beer styles (BA and CA), initially at 4 °C, were treated without filters (NF), with a clear filter, or with a bronze-tinted filter at four different residence times (5 min, 2.5 min, 0.5 min, and 0.25 min) using the experimental design summarized in Table 1. The temperature was monitored using a T-type flexible-wire thermo-couple (wire diameter 0.0762 mm, Omega Engineering Corp., Stamford, CT, USA) with the readings recorded at 1 s intervals using a data acquisition system (HP34970A, Hewlett, Packard, Loveland, CO, USA). The temperature rise was less than 5 °C for all treatments, with a maximum temperature rise of 4.8 °C reported for the 16.75 J/cm^2^ treatment without filters. Control samples were not treated with PL nor pumped through the system. The treated samples were collected into aluminum foil-wrapped glass bottles to block light transmission and approximately 30 mL of the treated sample was then poured into a 50 mL brown centrifuge tube, closed, and kept at −20 °C for 3-MBT quantitation. The remaining treated samples were used for other physicochemical analyses. Each treatment was performed in three biological replicates.
(1) Fluence was calculated as Fluence per flash×3 Hz×residence time

### 2.4. Culture Preparation and Enumeration

The *L. brevis*, CCC B1300, Molson strain was obtained from the MICB 421 culture collection at the University of British Columbia Department of Microbiology and Immunology, as referred by Pam, Razaei, and Soor [22]. The strain was grown in De Man, Rogosa, and Sharpe (MRS) broth (Sigma, Toronto, ON, Canada) and incubated in a sealed bottle at 30 °C overnight (first incubation). The incubated strain was streaked on MRS agar plates and incubated inside an anaerobic jar with CO_2_ generators (Becton, Dickinson and Company, Franklin Lakes, NJ, USA) for 48 h at 30 °C. The incubated MRS agar plates were stored at 4 °C as a backup.

The strain was acclimatized to each beer style to achieve a more realistic simulation of the contaminating strain within the breweries. Acclimatization induced the expression of *horA* genes that have been associated with the strain’s hop resistance capability [4]. The acclimatization procedure was adopted from Haakensen et al. [23]. An aliquot (1 mL) of the incubated unacclimatized *L. brevis* MRS broth was pipetted into 100 mL of sterilized 85:15 medium containing 85% beer and 15% MRS broth. Inoculated acclimatization mediums were incubated in a sealed media bottle for 48 h at 30 °C. BA and CA acclimatized strains were each streaked into three MRS agar plates and incubated inside an anaerobic jar with CO_2_ generators (Becton, Dickinson and Company, Franklin Lakes, NJ, USA) for 48 h at 30 °C. The agar plates were stored at 4 °C until needed.

Sterilized MRS broth (100 mL) was inoculated with 4–5 colonies of acclimatized strain using a sterile loop. The inoculated MRS broth (10 mL) was incubated overnight at 30 °C in a pre-autoclaved 1L media bottle with a stirring magnet and partially wrapped with aluminum foil with only the volume indicator lines exposed. The top lids of two 473 mL cans of beer, corresponding to the acclimatized strain, were flamed, and the contents were aseptically and slowly poured in at an angle to avoid excessive foaming. Another layer of aluminum foil was loosely wrapped around the 1 L bottle. The sample was homogenized on a magnetic stir plate for 1 min. The estimated total volume of 956 mL was used for one treatment. 

Untreated and treated samples were serially diluted in test tubes of 9 mL sterile 0.1% peptone water until 10^−6^ dilution was reached and then plated in triplicate on 3M LAB Petrifilms (Petrifilms). Loaded Petrifilms were incubated for 48–72 h at 30 °C. Colonies in Petrifilms with 30–300 CFU were enumerated. Petrifilms were used due to their convenience and accuracy. Kanagawa et al. [24] showed no significant differences in bacterial counts enumerated by Petrifilms and MRS agar medium.

### 2.5. Physicochemical Properties Analysis

Before measuring physicochemical properties, beers were degassed in aluminum foil-wrapped glass bottles with caps slightly unscrewed using an Elmasonic S 30H ultrasonicator (Elma Schmidbauer GmbH, Singen, Germany) at 37 kHz for 30 min.

An Accumet AE 150 digital pH meter (Fisher Scientific, Waltham, MA, USA), calibrated at pH 4, 7, and 10, was used to measure the pH of control and treated samples within an hour after treatment. Approximately 15 mL of sample was used per reading and discarded. The pH was measured in triplicate.

Control and treated samples were analyzed for color using the standard reference method (SRM) adopted from ASBC [25], expressed in dimensionless SRM values. Samples were prepared in 3 mL two-sided plastic cuvettes with a 10 mm light path. BA samples were prepared by backward-pipetting 3 mL of sample into the plastic cuvettes without dilution. CA samples were diluted at a 5:1 distilled water-to-beer ratio to obtain an absorbance reading below 1. Thus, the CA samples were prepared by backward-pipetting 2.5 mL of distilled water into plastic cuvettes followed by 0.5 mL of beer sample. Each sample’s absorbance was read at 430 nm using a UV-vis spectrophotometer (Shimadzu UV-1800 UV-Vis Spectrophotometer). Samples were prepared and measured in triplicate. The SRM value was obtained using Equation (2). The dilution factors used for BA and CA were 1 and 6, respectively.
(2)$$SRM\;value=abs_{430}\times dilution\;factor\times 12.7$$

The bitterness of control and treated samples for both beers was determined using a modified version of the manual isooctane extraction method adopted from ASBC [25]. For this purpose, 10 mL of each sample was transferred into an aluminum-foil-wrapped 50 mL clear polypropylene centrifuge tube (Corning, Corning, NY, USA) and acidified with hydrochloric acid (HCl, 1 mL, 3M). Afterward, 20 mL of isooctane was added to extract the bitterness compounds in the fume hood. The mixture was homogenized horizontally at 25 °C in a mechanical orbital shaker at 300 rpm for 15 min using the C76 Water Bath Shaker (New Brunswick Scientific, Enfield, CT, USA). The organic layer was separated from the aqueous layer by centrifuging at 400× *g* for 5 min at 20 °C using the Sorvall Legend X1R Centrifuge (Thermo Scientific, Waltham, MA, USA). The absorbance of the organic layer (upper) was measured spectrophotometrically at 275 nm by pipetting into 3 mL quartz cuvettes. The aluminum foil wrapping around the centrifuge tube was removed to provide visual assistance in the transferring of the organic layer. The absorbance was measured using the UV-1800 UV-Vis Spectrophotometer (Shimadzu, Kyoto, Japan) calibrated with acidified isooctane (20 mL isooctane with 1 mL 3M HCl added). The IBU was obtained by using Equation (3). Readings were performed in duplicate.
(3)$$IBU=abs_{275}\times 50$$

TSS in control and treated samples were measured using a handheld 0–32% Brix refractometer (Sper Scientific, Scottsdale, AZ, USA), and expressed in °Brix. TSS were measured in triplicate.

### 2.6. Headspace SPME GC/MS Analysis of 3-MBT Content

In order to analyze the 3-MBT content, volatiles from the headspace of 5 mL of the sample (in 20 mL glass SPME vials with 10 µL ethyl butyrate solution) were adsorbed on a divinylbenzene/carboxen/polydimethylsiloxane (DVB/CAR/PDMS) solid-phase micro-extraction (SPME) fiber (50/30 μm, 1 cm, Sigma Aldrich, Oakville, ON, Canada), according to the protocols of Singh et al. [26]. The prepared samples were homogenized by vortexing for 10 s, followed by preincubation without the fiber in the oven of an SPME autosampler (GC Sampler 80, Agilent Technologies, Santa Clara, CA, USA) connected to a 7890A gas chromatograph (GC) coupled to a 5975C single quadrupole mass spectrometer (MS) detector (Agilent Technologies, USA) for 5 min at 45 °C while agitated at 300 rpm. The fiber was exposed to the sample headspace for 30 min at 45 °C while agitated at the same speed. Fibers were conditioned at 250 °C for 20 min after each analysis to prevent the cross-contamination of non-targeted semi-volatile compounds. Volatiles were thermally desorbed from the SPME fiber in the GC injection port at 250 °C for 3 min, and then separated on a polar DB-Wax column (100% polyethylene glycol, 30 m × 0.25 mm ID × 0.25 μm, J&W Scientific, Folsom, CA, USA) with helium as carrier gas at a flow rate of 0.9 mL/min. Chromatograph conditions were optimized for the analysis of 3-MBT but allowed the other non-targeted compounds to elute. The GC oven temperature was programmed at 35 °C for 4.36 min, followed by an increase of 15 °C/min until 230 °C, which was maintained for 5 min. The total GC run time was 22 min. The interface, ion source, and quadrupole temperatures were 250 °C, 230 °C, and 150 °C, respectively. Eluted compounds were simultaneously detected in SCAN (mass range of 35–500 *m*/*z*) and Selected Ion Monitoring (SIM) modes, operating in electron impact ionization mode at 70 eV. The SIM-targeted detection of 3-MBT was performed by tracking the ions 69 (target ion), 41 (qualifier ion 1), and 102 (qualifier ion 2). 3-MBT was identified on the chromatograph by comparing its retention times and accurate mass spectra with standards (Aroxa, Surrey, UK) and using the Mass Spectra Library (NIST MS Search 2.2 and MS Interpreter) on Chemstation software (version E.02.02.1431, Agilent Technologies, Santa Clara, CA, USA). Treatments were analyzed in triplicate.

Standards were obtained in the form of capsules containing 100 ng 3-MBT per capsule. The standard was dissolved; serially diluted in 5% ethanol (to mimic the matrix of beer) to concentration levels of 2000, 1000, 500, 250, and 125 ng/L; and similarly analyzed through SPME-GC/MS to create the calibration curve. This calibration curve was employed for the quantification of 3-MBT in control and treated beer samples, and the concentrations were corrected for the matrix effect %, the latter being calculated following Equation (4).
(4)$$\%\;Matrix\;effect=\left(\frac{\mathrm{area}\;\mathrm{spiked}\;\mathrm{sample}-\mathrm{area}\;\mathrm{non}-\mathrm{spiked}\;\mathrm{sample}}{\mathrm{area}\;\mathrm{standard}\;\mathrm{in}\;\mathrm{solution}}-1\right)\;\times \;100$$

To ensure that the identified 3-MBT in the different beers were above their limits of detection (LOD) and the quantified 3-MBT were above their limits of quantification (LOQ), the sensitivity was assessed in terms of LODs and LOQs as three and ten times the signal-to-noise ratio (S/N), respectively. For linearity, the coefficient of determination (R^2^) was subsequently calculated, and the linear range was also established.

### 2.7. Statistical Analysis

One-way analysis of variance (ANOVA) and least significant difference (LSD) were used at α = 0.05 for microbial, color, bitterness, pH, and TSS analyses, while two-way ANOVA and LSD were used at α = 0.05 to compare concentrations of 3-MBT using Minitab (Version 21.3.0, Minitab LLC, State College, PA, USA). A linear correlation analysis was conducted on 3-MBT calibration curves using Microsoft Excel (Microsoft Excel for Mac Version 16.69, Microsoft Corporation, Redmon, WA, USA).

## 3. Results

### 3.1. L. brevis Reduction in Beer Processed Using Pulsed Light with/without UV Filters

Isolated *L. brevis* streaked on MRS agar plates display off-white circular, raised colonies with entire margins, extending to 1–2 mm in diameter. *L. brevis* colonies grown on Petrifilms were dark red, and a majority had irregular forms. (Appendix A). The microbial reduction in *L. brevis* in samples treated without filters for a maximum (5 min) treatment time reached up to 4.2 log reductions for BA from an initial 7.7 log CFU/mL and up to 2.4 log reductions for CA from 7.0 log CFU/mL. With a clear filter and BA, the maximum microbial reductions were limited to 1.2 log reductions, while with a bronze-tinted filter, only a maximum of 0.8 log reductions could be obtained with a 5 min treatment time. Significant differences between BA and CA were only observed for 2.5 and 5 min of treatment with no filters, where BA displayed higher log reductions, while for other filters at all treatment times, no significant differences existed. More than 0.5 log reductions were obtained with a clear filter and bronze-tinted filter only for a maximum residence time of 5 min (Figure 2).

### 3.2. Concentration of 3-MBT in Beer Processed Using Pulsed Light with/without UV Filters

3-MBT concentrations in samples were quantified based on the abundance of ion 102 due to the interference of another compound that fragged ions with 69 *m*/*z* in no-filter-treated samples, which was not observed in control or filter-treated samples. The Kovats Index (KI) of 3-MBT in BA and CA was approximately 1053, matching with the PubChem database for polar columns. The peak corresponding to 3-MBT was clearly visible in samples treated without filters (Appendix A), and the mass spectrum of the 3-MBT peak in beer matched the mass spectrum of the standard in solvent (Appendix A). The calibration curve for ion 102 was linear (Appendix A). The limit of detection (LOD) and limit of quantification (LOQ) of ion 102 in solvent were 19.55 ng/L and 64.50 ng/L, respectively, while the matrix effect for ion 102 was −4.1% for CA and 52.1% for BA. Although 3-MBT was detected in all the samples, all samples treated with a filter, except for the BA treated for 5 min with a clear filter, did not have quantifiable concentrations of 3-MBT (Table 2), i.e., the MS signals of the samples were <LOQ. The BA samples treated without filters had significantly higher 3-MBT concentrations (*p < 0.05*) than CA samples treated similarly; specifically, BA treated for 0.25 min exceeded the 3-MBT concentration in CA treated for 5 min. The rate of 3-MBT formation was observed to begin plateauing after 0.5 min.

### 3.3. Physicochemical Properties of Beer Processed Using Pulsed Light with/without UV Filters

Table 3 summarizes the selected physicochemical properties of BA and CA beer samples. CA beers had an overall higher SRM than BA beers. Control BA had significantly higher SRM than all the treated BA samples (*p <* 0.05), and there were significant but small differences (*p <* 0.05) among the color of the treated BA samples. Specifically, BA samples treated with no filter for 5 min had the lowest SRM, indicating that it became lighter in color with PL treatment, while the CA beer treated for 5 min and with no filter had the highest SRM, indicating that it became darker in color with PL treatment. Beers treated with a bronze-tinted filter had the lowest color change, followed by those treated with a clear filter, and maximum color change was observed with those treated without filters.

Control BA and CA had IBU values of 23.0 and 41.4, respectively. The IBU of control BA was significantly higher (*p <* 0.05) than its treated counterparts, while that of CA was significantly lower (*p <* 0.05) than treated samples. Among treated samples, the mean IBU of BA samples had an overall decreasing trend as fluence increased, while that of CA had an overall increasing trend with the increase in fluence.

The pH levels of CA samples were all higher than that of BA samples. The pH of the control BA was significantly higher (*p <* 0.05) than its treated counterparts. The pH levels of control CA were significantly higher (*p <* 0.05) than the other treated CAs, except for clear-filter and bronze-tinted-filter-treated CAs for 0.25 min. PL treatment with higher residence times tended to decrease the pH of both beers.

The TSS in BA beers were relatively higher compared to that of CA beers. For BA samples, all PL treatments, except clear and bronze-tinted filters at 0.25 min residence times, generally resulted in a significant decrease in TSS (*p <* 0.05), with no-filter treatment at 5 min producing the lowest TSS. For CA samples, only PL treatments greater than 2.5 min residence time produced significant differences (*p <* 0.05) in TSS for bronze-tinted-filter and no-filter treatments, whereas no significant differences were produced with clear filters.

Overall, PL treatments applied at higher residence times with no filter produced larger color changes, lower pH, and lower TSS in both beer samples, whereas PL also decreased the bitterness of BA and increased the bitterness of CA beers. It must be noted that the PL treatment with no filter for 5 min exceeded the maximum fluence regulations of 12 J/cm^2^ PL treatment and may not be applicable to food.

## 4. Discussion

The PL treatment of beers without filters allowed exposure to the broadest spectrum of electromagnetic radiation from 200 to 1100 nm, including UV, visible light, and IR rays, resulting in 4.2 and 2.4 log reductions, respectively, for BA and CA beers for 5 min of treatment. However, this treatment without filters imparted 16.8 J/cm^2^, exceeding the fluence regulatory limits for food and beverage. Following this, at 8.4 J/cm^2^ for 2.5 min residence time, PL treatment without a filter resulted in a maximum usable 2.7 and 1.6 log reductions for BA and CA beers, respectively. Treatment at this intensity resulted in the generation of 1629 ng/L 3-MBT, as well as significant changes in color, bitterness, pH, and TSS. Generally, higher fluence was needed to achieve higher and more practical microbial reductions. The microbial deactivation of beers treated without filters at 2.5 min and 5 min residence time (8.4 J/cm^2^) was comparable to the study conducted by Hosseini et al. [16], where they observed more than three log reductions for *E. coli* in non-alcoholic beer using 266 nm-wavelength lasers for 10 J/cm^2^. However, the differences in reductions could be due to the different wavelengths used, the penetration depth of the beverages treated, the bacteria targeted, and liquid volume per cm^2^ of surface exposed to PL.

The application of a clear filter, for 5 min residence time (8.9 J/cm^2^ fluence, only marginally larger than the 2.5 min residence time treatment without a filter) resulted in 1.2 (BA) and 1.0 (CA) log reduction in *L. brevis* but led to minimal 3-MBT formation and fewer significant changes in physicochemical parameters as compared to 2.5 min of treatment without a filter. The application of a bronze-tinted filter resulted in fewer than 1.0 log reductions of *L. brevis* at all treatment fluences tested (0.1–1.3 J/cm^2^), with non-quantifiable 3-MBT formation and fewer significant changes in physicochemical properties as compared to a clear filter. Mandal et al. [7] highlighted the importance of the UV-C (200–280 nm) portion of the spectrum for microbial deactivation. As both the clear and bronze-tinted filters blocked UV radiation, treatments with filters underwent significantly less (*p* < 0.05) or insignificantly different (*p* > 0.05) microbial reductions in the same beer style and residence time as compared to treatment without filters. With the bronze-tinted filter, as further areas of visible light and IR rays were removed, resulting in less than 1/6th of the fluence of a clear filter, the decontamination power was significantly reduced. It must be mentioned that a maximum tested fluence of 1.3 J/cm^2^ is considerably low, and achieving higher decontamination with a longer residence time cannot be discarded. Moreover, the temperature rise was recorded to be less than 5 °C in all cases with or without filters at all residence times, suggesting a limited increase in temperature due to IR’s thermal effects, possibly due to the mixing of the liquids. Although treatments with both filters had reduced deactivation effects in our study, both BA and CA beers still had recognizable deactivation power contributed by visible light and IR rays; deactivation was possibly achieved due to the photothermal effect of IR rays and the phototoxic effect of inducing high levels of bacterial-produced ROS from intense visible light [7,12,27].

Significantly higher (*p <* 0.05) microbial reduction, the generation of 3-MBT, and changes in physicochemical parameters were observed in BA beers compared to CA beers, which was attributed to the contribution of increased light penetration depth in the lighter-colored BA, as shown with overall lower SRM values. Penetration depth is influenced by light absorbance, the inverse of transmittance [28]. Fluids of lighter color have lower absorbance and higher transmission in the visible light spectrum [29]. The truncated penetration depth of UV rays in CA could have been due to higher levels of melanoidins in darker beers [30]. Melanoidins have enough absorbance in the visible spectrum for humans to perceive the darkness in dark beer, but they have especially higher absorbance in the UV spectrum [31]. Kang [31] observed that higher intensities of melanoidins had increased absorbance in the UV region, which is synonymous with the lower UV penetration depth and lower microbial reduction in CA. High levels of melanoidins in CA beers would have been sourced from malts added during its production.

As CA has higher concentrations of iso-α-acids than BA, indicated by higher IBU, the former theoretically would have a higher concentration of reactants to degrade and form more 3-MBT than the latter. However, the opposite trend was observed, possibly due to the penetration depths of both beers. High levels of melanoidins in the dark-colored CA, potentially reaching 1.49 g/L for dark beers, would have absorbed a substantial amount of UV rays compared to riboflavin and iso-α-acids, usually present in 0.2–0.4 mg/L and 17–55 mg/L, respectively [3,32,33]. Furthermore, melanoidins are large, heterogeneous polymers with high molecular weights between 85 and 232 kDa [34]. Hence, melanoidins are bigger in size and could shadow riboflavin and iso-α-acids from the UV incidence, preventing the “lightstruck” reaction. While only slight microbial deactivations were displayed in BA treated without filters for 0.25 and 0.5 min residence times, substantial concentrations of 3-MBT were still formed, suggesting that the energy threshold to form 3-MBT was lower than to deactivate *L. brevis*.

The high penetration depth characterized by PL in BA samples allowed the energy carried in electromagnetic radiation to be dispersed more evenly throughout the coil chamber’s diameter. With low fluences, the energy absorbed by each thin radial layer of the fluid in the coil chamber was not sufficient to deactivate *L. brevis*, resulting in overall less microbial reduction. On the other hand, the low penetration depth of CA caused less energy dispersion but allowed energy to be absorbed and concentrated on the outermost fluid layer nearing the chamber coil’s interior surface. As fluids were treated under laminar conditions with Reynolds numbers ranging from 154 to 2053, parabolic velocity profiles formed in the coil chamber during treatment. The parabolic velocity profile endorses maximum fluid velocity in the coil’s centerline. In contrast, the fluid velocity is lower near the surface, further enhancing surface decontamination in CA as it concentrates the electromagnetic energy [17,35]. Fluid entrance regions were between 0.083 and 1.1 m, which allowed fully developed fluid flow during a large portion of each treatment. Considering particles dispersed throughout the cross-section of the fluid, there is a possibility of unequal treatment between all radial distances from the centerline inside the fully developed flow.

3-MBT formation was quantified in beers treated without filters at all residence times and only at the highest residence time (8.9 J/cm^2^ fluence) with a clear filter. As the residence time of beer treated without filters increased, there was a substantial increase in the final concentration of 3-MBT due to the increased fluence that drove the “lightstruck” reaction forward. This finding was aligned with Masuda et al. [36], who quantified more 3-MBT in beers exposed to more sunlight. BA and CA beers treated without a filter, even for the shortest residence time, resulted in concentrations of 3-MBT 11.77 and 3.693 times higher than the LOQ, respectively, while 3-MBT in beers treated with the filters even for the longest residence time (except BA PL-treated with a clear filter at a maximum of 5 min residence time) stayed below the LOQ. This indicated that UV rays removed by the clear and bronze-tinted filters were needed to substantially form 3-MBT in beers. Since the higher transmission of visible and infrared light by treating with the clear filter instead of the bronze-tinted filter in BA allowed some 3-MBT formation, it is also elucidated that some other visible wavelengths, especially those in the 400–500 nm range, can induce 3-MBT formation. This observation aligns with the observation of Kuroiwa et al. [18], who quantified high 3-MBT formation in beer exposed to 500 nm light [20,21]. For the control and all filter-treated beer samples except one (clear filter at maximum 8.9 J/cm^2^ fluence), 3-MBT was detected, but the concentrations were <LOQ. Therefore, the clear and bronze-tinted filters worked to prevent 3-MBT formation during PL processing. This observation indicated that the reaction to form 3-MBT was highly sensitive to UV rays below 400 nm.

Color analysis showed that CA beers were darker than BA beers due to the addition of malts. The data displayed in Table 3 showed that BAs were turned lighter when exposed to higher fluences, especially observable after treatment without filters for 2.5 and 5 min, suggesting that PL treatment may have affected the light degradation properties of the colored compounds in beer, including melanoidins and polyphenols [30]. This pattern was supported by the UV-induced discoloration and degradation of melanoidins observed by Yilmaz and Findik (2016), where longer exposure to UV rays increased discoloration. Discoloration may have also been achieved through other mechanisms working simultaneously to alter the color of beer without UV exposure, which requires more specific research. For CA beers, it was observed that UV exposure increased the SRM value, as UV could also trigger the polymerization of phenol compounds contributing to increased darkness [37,38].

The bitterness of CA beers was seen to increase with fluence, while that of BA beers was seen to decrease with fluence. The decrease in BA bitterness from PL treatment is indicative of the photo-oxidation and degradation of iso-α-acids influenced by UV, which supports the formation of 3-MBT [19,39]. However, the opposite trend for CA beers is interesting. Since the manual isooctane extraction method also measures non-isomerized-α-acids and oxidative polar compounds, some of these might have been formed due to the malt-derived polar compounds in CA [40]. The corresponding change in bitterness was lower with filters, with generally no significant differences between the two filters, suggesting that the application of most of the bitterness change was mostly UV-derived. While instrumental bitterness is known to differ from perceived bitterness, higher bitterness is generally associated with a negative involuntary perception [41], which would require sensory trials to further investigate the effect of PL on perceived bitterness in different beer types.

There were overall small reductions in the pH after PL treatment in both beers. Overall, the pH decreased as fluence increased and the filter settings changed from bronze-tinted filter to clear filter to no filter. The mixing with oxygen in the coil chamber and the PL treatment may have caused photooxidation in compounds such as terpenes, majorly identified in hops essential oils carried into beer [26]. The oxidation of terpenes leads to carbonyl production such as organic acids, thus slightly lowering the pH [42].

The bulk of TSS measured in finished beer accounts for residual sugar unfermented by yeast. Differences in residual sugars could potentially alter the perceived bitterness of beer [43]. Although there were significant but minor differences (*p <* 0.05) between TSS, the impact on the perceived bitterness was not tested. TSS measured in each beer did not have an observable trend that correlated with fluence or wavelength exclusions. As the beers tested herein were unfiltered and unpasteurized before canning, unequal amounts of suspended yeast may still have been present [44]. Residual yeast can continue to ferment very small amounts of fermentable sugar, resulting in a slight variation of residual sugars, as measured by TSS, during analysis.

Overall, there were many insignificant differences (*p >* 0.05) in color, bitterness, pH, and TSS between treatments in each beer, suggesting that PL treatments do not impose radical measurable changes in the selected physicochemical properties of the beers tested. However, sensory analysis to correlate instrumental differences with sensory differences need to be conducted to support this statement. Since there was no previous research carried out specifically on the application of PL processing on beer, this study sheds light on a wide variety of future research that could be conducted to inhibit the shortcomings while increasing the desired impacts of PL. Toxicological tests should also be conducted on PL-treated beers to confirm compounds generated during the process would not impact human health differently from untreated beer.

## 5. Conclusions

This study showed that both clear and bronze-tinted PSS are effective optical filters to prevent UV transmittance. PL treatment resulted in minute differences in the color, bitterness, pH, and TSS of each beer, though several treatments showed significant differences (*p <* 0.05) in each physicochemical property. However, the outcomes of PL processing showed differences between different styles/colors of beer. For example, the bitterness of bitter CA beer increased with PL treatment, while that of lighter BA beer decreased. The research suggested that while PL treatment without filters was effective in achieving microbial reductions, they caused a significant “lightstruck” phenomenon due to 3-MBT formation. The use of UV filters was effective in controlling the 3-MBT formation, but with limited microbial reductions. Further research is needed to optimize the PL treatment and filter wavelengths to achieve practical microbial reduction while minimizing the formation of 3-MBT. Lower fluence PL treatment could also be combined with other non-thermal approaches, such as ultrasonication, to minimize 3-MBT formation, thus adding another hurdle for microbial survival.

## Figures and Tables

**Figure 1 foods-12-00684-f001:**
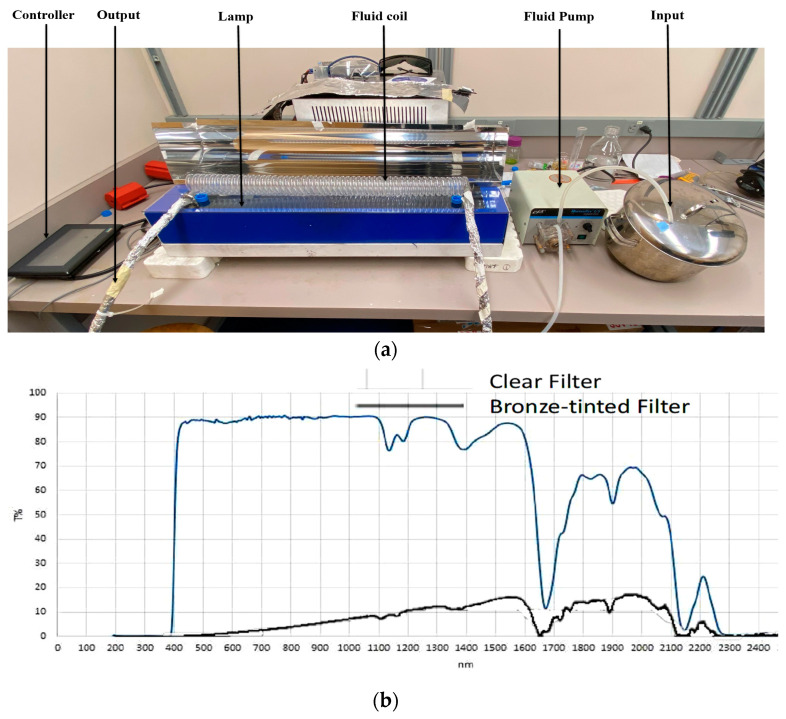
(**a**) Pilot-scale pulsed light operation set-up; (**b**) transmission spectrum of 3 mm clear and bronze-tinted polycarbonate solid sheets (lighter blue line for clear filter; darker black line for bronze-tinted filter).

**Figure 2 foods-12-00684-f002:**
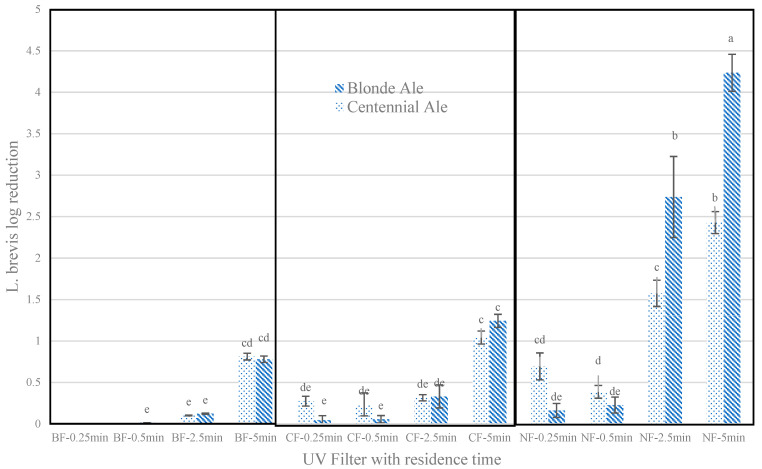
*L. brevis* log reductions after different PL treatments (bar values represent mean values, and error bars represent standard deviation, *n* = 3). One-way analysis of variance (ANOVA) was conducted on log reductions across all treatments. The same letters on top of bars indicate that values are not significantly different (*p >* 0.05). Treatments are reported as (filter condition)-(residence time (s)). NF: no filter; CF: clear filter; BF: bronze-tinted filter; UV: ultraviolet.

**Table 1 foods-12-00684-t001:** Accumulated fluence (J/cm^2^) for different treatments imparted on the two types of beer in this study.

Filter	Residence Time (min)			
	5	2.5	0.5	0.25
	Fluence (J/cm^2^) ^1^			
No filter	16.75	8.37	1.67	0.84
Clear filter	8.93	4.46	0.89	0.45
Bronze-tinted filter	1.32	0.66	0.13	0.07

^1^ Equation (1).

**Table 2 foods-12-00684-t002:** Concentration of 3-MBT (3-methylbut-2-ene-1-thiol) in control and pulsed-light-treated beers.

Filter	Beer Style	Residence Time (min)	Concentration of 3-MBT (ng/L) *
Control	Blonde ale (BA)	0.0	<LOQ
	Centennial red ale (CA)	0.0	<LOQ
No filter	BA	5	1832 ± 149 ^ax^
		2.5	1629 ± 156 ^ax^
		0.5	1521.9 ± 320.1 ^ax^
		0.25	759.4 ± 251.5 ^bx^
	CA	5	682.4 ± 81.2 ^ay^
		2.5	548.1 ± 114.8 ^ay^
		0.5	469.6 ± 94.1^aby^
		0.25	238.2 ± 24.6 ^by^
Clear filter	BA	5	121 ± 39 ^z^
		2.5	<LOQ
		0.5	<LOQ
		0.25	<LOQ
	CA	5	<LOQ
		2.5	<LOQ
		0.5	<LOQ
		0.25	<LOQ
Bronze-tinted filter	BA	5	<LOQ
		2.5	<LOQ
		0.5	<LOQ
		0.25	<LOQ
	CA	5	<LOQ
		2.5	<LOQ
		0.5	<LOQ
		0.25	<LOQ

* Data are shown as mean ± standard deviation (*n* = 3). Two-way analysis of variance (ANOVA) was conducted for values above the LOQ. Letters a and b reported after values indicate significant differences between treatments of each beer style (*p <* 0.05). Letters x–z reported after values indicate significant differences between beer styles (*p <* 0.05). LOQ: limit of quantification; 3-MBT: 3-methylbut-2-ene-1-thiol.

**Table 3 foods-12-00684-t003:** Color (SRM), bitterness (IBU), pH, and total soluble solids (TSS) of control and pulsed-light-treated blonde ale (BA) and centennial red ale (CA) beers.

Filter	Residence Time (min)	SRM *	IBU *	pH *	TSS (°Brix) *
Blonde Ale (BA)					
Control	0	4.12 ± 0.14 ^a^	23.0 ± 0.6 ^a^	4.09 ± 0.01 ^a^	6.2 ± 0.1 ^a^
No filter	5	3.35 ± 0.21 ^d^	17.1 ± 0.9 ^d^	4.03 ± 0.03 ^bc^	5.8 ± 0.2 ^b^
	2.5	3.50 ± 0.15 ^c^	18.6 ± 1.0 ^c^	4.05 ± 0.02 ^b^	5.9 ± 0.1 ^b^
	0.5	3.65±0.09 ^bc^	19.6 ± 1.4 ^bc^	4.04 ± 0.02 ^bc^	5.9 ± 0.0 ^b^
	0.25	3.59 ± 0.05 ^c^	20.4 ± 1.1 ^b^	4.04 ± 0.03 ^bc^	5.9 ± 0.1 ^b^
Clear filter	5	3.61±0.08 ^bc^	19.9 ± 0.9 ^b^	4.04 ± 0.01 ^c^	5.9 ± 0.1 ^b^
	2.5	3.67± 0.07 ^bc^	20.3 ± 0.6 ^b^	4.06 ± 0.04 ^b^	5.9 ± 0.1 ^b^
	0.5	3.62 ± 0.11 ^c^	20.7 ± 0.9 ^b^	4.04 ± 0.01 ^c^	5.9 ± 0.1 ^b^
	0.25	3.60 ± 0.06 ^c^	21.5 ± 0.7 ^b^	4.05 ± 0.02 ^bc^	5.9 ± 0.3 ^ab^
Bronze-tinted filter	5	3.71±0.13 ^b^	19.1 ± 1.2 ^c^	4.04 ± 0.01 ^c^	5.9 ± 0.2 ^b^
	2.5	3.70 ± 0.10 ^b^	19.5 ± 1.3 ^bc^	4.05 ± 0.01^b^	5.8 ± 0.2 ^b^
	0.5	3.72 ± 0.03 ^b^	21.7 ± 1.2 ^b^	4.05 ± 0.02 ^bc^	5.9 ± 0.1 ^b^
	0.25	3.56 ± 0.12 ^c^	20.9 ± 0.5 ^b^	4.05 ± 0.02 ^b^	6.0 ± 0.3 ^ab^
Centennial Red Ale (CA)					
Control	0.0	39.0 ± 1.7 ^a^	41.4 ± 4.2 ^b^	4.20 ± 0.01 ^a^	5.7 ± 0.1 ^a^
No filter	5	48.6 ± 1.1 ^c^	46.5 ± 3.2 ^a^	4.15 ± 0.00 ^cd^	5.3 ± 0.2 ^c^
	2.5	44.6 ± 1.1 ^b^	42.7 ± 3.2 ^ab^	4.16 ± 0.00 ^cd^	5.4 ± 0.2 ^bc^
	0.5	40.5 ± 0.8 ^a^	45.2 ± 0.5 ^a^	4.16 ± 0.01 ^bc^	5.6 ± 0.1 ^ab^
	0.25	39.4 ± 0.6 ^a^	39.1 ± 2.9 ^b^	4.16 ± 0.01 ^cd^	5.6 ± 0.1 ^ab^
Clear filter	5	43.4 ± 2.1 ^b^	45.3 ± 0.5 ^a^	4.15 ± 0.01 ^b^	5.6 ± 0.1 ^ab^
	2.5	41.6 ± 0.7 ^ab^	43.2 ± 2.0 ^ab^	4.16 ± 0.01 ^c^	5.7 ± 0.1 ^a^
	0.5	42.0 ± 1.3 ^ab^	40.6 ± 2.3 ^b^	4.13 ± 0.02 ^e^	5.6 ± 0.2 ^ab^
	0.25	38.1 ± 0.9 ^a^	41.5 ± 1.3 ^b^	4.20 ± 0.01 ^a^	5.6 ± 0.1 ^ab^
Bronze-tinted filter	5	42.5 ± 2.1 ^ab^	45.1 ± 0.5 ^a^	4.14 ± 0.01 ^b^	5.5 ± 0.1 ^b^
	2.5	41.6 ± 2.1 ^ab^	44.8 ± 0.5 ^a^	4.17 ± 0.01 ^b^	5.5 ± 0.1 ^b^
	0.5	41.3 ± 1.9 ^ab^	41.9 ± 1.7 ^b^	4.15 ± 0.01 ^d^	5.6 ± 0.1 ^ab^
	0.25	39.8 ± 0.5 ^a^	42.7 ± 2.0 ^ab^	4.17 ± 0.01 ^b^	5.6 ± 0.1^ab^

* Data are shown as mean ± standard deviation (*n* = 9). Four separate one-way analyses of variance (ANOVAs) were conducted for each physicochemical property. The same letters reported after values indicate no significant differences in respective physicochemical properties (*p >* 0.05). SRM: standard reference method; IBU: international bitterness unit; TSS: total soluble solids.

## Data Availability

All data are available in this manuscript or through a request to the corresponding author.

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
