# Peer review of "Effect of UV Filters during the Application of Pulsed Light to Reduce Lactobacillus brevis Contamination and 3-Methylbut-2-ene-1-thiol Formation While Preserving the Physicochemical Attributes of Blonde Ale and Centennial Red Ale Beers"

_foods, 2023, doi:10.3390/foods12040684_

Round 1

Reviewer 1 Report

Major remarks.

Line 73. Fluence is energy received by the product SURFACE per unit area. Energy received by the product SURFACE per unit area per unit time is fluence rate. Overall, it seems that there is a confusion in the use of the term “fluence” and its different derivatives. “Accumulated fluence” (table 1) does not exist although it has been used sometimes, it is just fluence (indeed fluence accumulates with each flash, but the whole of it is just fluence); “fluence received” is redundant since fluence is already the energy received by the surface.

Lines 161-166. How was light transmission measured?

Line 168. How was fluence measured? Which wavelengths? At the proximal or the distal end of the reactor? The latter is extremely important since it is not the same two treatment conditions using the same fluence but with different spectral composition.

Lines 208-211. It is not clear why media was inoculated and then autoclaved?

Line 434. Why temperature was not controlled? How was the temperature change during treatments? IR means heat, so if IR is related to microbial inactivation, the beer was heated during treatments.

Line 441. I would write that those liquids have lower absorbance not that they tend to have lower absorbance.

Lines 551-554. This conclusion is not derived from a previous discussion nor related to the goal of the manuscript.

General comment. There are many authors who have published about pulsed light technology, so please reduce self-citations. Furthermore, using self-citation to compare results and support conditions is not as suitable as citing others, since you can be biased (this is not a personal comment, I do not refer specifically to you, any researcher can be biased towards his/her own conclusions).

Conclusions. I missed a definitive conclusion about the best fluence/optical filter combination. Which one recommend you and why and which is its practical significance? For example, if 1.3 J/cm2 and bronze-tinted filter is the best according to you, it has only 1 log inactivation, is that of practical significance? If you believe that you should not recommend a specific combination as final, which one and why would you select for further research as referred in lines 559-563?

Minor remarks.

Line 18 and elsewhere. 1400 nm seems too much. For example, in lines 136 and 403 you wrote 1100 nm. Otherwise, cite references in which the emission spectrum of the employed lamp reached that wavelength.

Line 53. Write Gram with first uppercase letter.

Figure S1. The picture of the beers in cups should be more professional, with an uniform illumination and background.

Tables S1 and S2. What is the symbol @?

Line 133. Correct fittewith.

Figure 1. Is it possible to use the same scale for figs. 1b and 1c.

Line 169. Standard deviation of the latter fluence is missed.

Line 179 and elsewhere. Controlled samples or control samples?

Table 1. The equation should not be place in the table, it may be relocated in the text if needed.

Line 203. “tween” or “Tween”? Which kind of Tween? Source?

Section 2.3. should appear before section 2.2. since bacteria were first cultured and then treated.

Line 230. Frequency of water change should be omitted.

Lines 231-234. Calibration and cleaning details should be omitted as well as the comment on stable reading.

Line 246. Calibration detail should be deleted.

Lines 269-273 Details such as “Before reading, the glass on the refractometer was rinsed with distilled water and wiped with Kimwipes”, “2-3 drops of the sample were transferred on the refractometer glass using a dropper”, “the plastic cover was closed, and °Brix was read through the eyepiece” should be deleted since they are widely known.

Line 280. Vertexing?

Figure 2. Negative Y-axis values must be deleted. It seems more logic to show data ordered by increasing time.

Line 413 and elsewhere. Use italics for scientific names.

Lines 458-460. This sentence can be omitted since it follows the same logic than the previous one.

Author Response

Major remarks.

Line 73. Fluence is energy received by the product SURFACE per unit area. Energy received by the product SURFACE per unit area per unit time is fluence rate. Overall, it seems that there is a confusion in the use of the term “fluence” and its different derivatives. “Accumulated fluence” (table 1) does not exist although it has been used sometimes, it is just fluence (indeed fluence accumulates with each flash, but the whole of it is just fluence); “fluence received” is redundant since fluence is already the energy received by the surface.

REPLY: WE AGREE WITH THE REVIEWER AND THE TERM IS USED CORRECTLY. HOWEVER, WE HAVE STILL USED ACCUMULATED FLUENCE, AS THE TERM FLUENCE IS GENERALLY DEFINED FOR STATIONARY OBJECTS, BUT FOR THE CONTINUOUS SYTEM, THIS IS THE FLUENCE ESTIMATED TO BE ACCUMULATED ACROSS THE RESIDENCE TIME, AND NOT ACROSS THE SURFACE.

Lines 161-168. How was light transmission measured?How was fluence measured? Which wavelengths? At the proximal or the distal end of the reactor? The latter is extremely important since it is not the same two treatment conditions using the same fluence but with different spectral composition.

REPLY: THE FOLLOWING HAS BEEN ADDED TO SPECIFY THE DEVICES USED TO MEASURE THE ENERGY TRANSMISSION:  The transmission through the filters, well as the fluence through the reactors, was measured using a 1 cm2 aperture polyelectric head sensor (PE80BF-DIF-C, Ophir-Spiricon LLC, UT, United states) with Nova II display (OphirSpiricon LLC, UT, United states) at the proximal end of the reactor, as described by Mandal and Pratap-Singh [17].

Lines 208-211. It is not clear why media was inoculated and then autoclaved?

REPLY: WE APOLOGIZE FOR THE CONFUSION, THE 1L MEDIA BOTTLES WERE FIRST INOCULATED AND THE INOCULATED MEDIA WAS THEN TRANSFERRED TO THEM FOR INCUBATION. WE HAVE REWRITTEN THE CONCERNED PARAGRAPH FOR CLARITY: "Sterilized MRS broth (100 mL) was inoculated with 4-5 colonies of acclimatized strain using a sterile loop. The inoculated MRS broth (10 mL) was incubated overnight at 30℃ in pre-autoclaved 1L media bottle with a stirring magnet and partially wrapped with aluminum foil with only the volume indicator lines exposed. The top lids of two 473 mL cans of beers, corresponding to the acclimatized strain, were flamed, and the contents were aseptically poured in slowly at an angle to avoid excessive foaming. Another layer of aluminum foil was loosely wrapped around the 1 L bottle. The sample was homogenized on a magnetic stir plate for 1 minute. The estimated total volume of 956 mL was used for one treatment."

Line 434. Why temperature was not controlled? How was the temperature change during treatments? IR means heat, so if IR is related to microbial inactivation, the beer was heated during treatments.

IT WAS NOT POSSIBLE TO CONTROL THE TEMPERATURE, BUT THE TEMPERATURE INCREASE WAS RECORDED TO BE LESS THAN 5oC. WE HAVE ADDED THIS INFO ON LINES:
"Also, temperature rise was recorded to be less than 5oC in all cases with or without filters at all residence times, suggesting limited increase in temperature due to IR’s thermal effects, possibly due to mixing of the liquids"

Line 441. I would write that those liquids have lower absorbance not that they tend to have lower absorbance.

REPLY: CORRECTED ON LINE 456. DETED THE WORD "Tend To"

Lines 551-554. This conclusion is not derived from a previous discussion nor related to the goal of the manuscript.

REPLY: DELETED THE LINES, "Hence, clear PSS could be implemented to display transparent-bottled beer displayed in-store coolers exposed to sunlight. These sheets are cost-efficient and durable materials such that breweries can install roof skylights on roofs to conserve energy, allowing sunlight to enter without UV to expose the beers inside. "

General comment. There are many authors who have published about pulsed light technology, so please reduce self-citations. Furthermore, using self-citation to compare results and support conditions is not as suitable as citing others, since you can be biased (this is not a personal comment, I do not refer specifically to you, any researcher can be biased towards his/her own conclusions).

REPLY: Deleted 2 self-citations, and replaced with work of other authors in 3 places. We apologize for this as this is the artifact of greater reliance on institutional knowledge by the students, and the point is well taken.

Conclusions. I missed a definitive conclusion about the best fluence/optical filter combination. Which one recommend you and why and which is its practical significance? For example, if 1.3 J/cm2 and bronze-tinted filter is the best according to you, it has only 1 log inactivation, is that of practical significance? If you believe that you should not recommend a specific combination as final, which one and why would you select for further research as referred in lines 559-563?

REPLY: ADDED THE FOLLOWING SPECIFIC CONCLUSION: "The research suggested that while PL treatment without filters was effective in achieving microbial reductions, they caused significant ‘lightstruck’ phenomenon due to 3-MBT formation. Use of UV filters were effective in controlling 3-MBT formation, but with limited microbial reductions. Further research is needed to optimize the PL treatment and filter wavelengths to achieve practical microbial reduction while minimizing the formation of 3-MBT."

Minor remarks.

Line 18 and elsewhere. 1400 nm seems too much. For example, in lines 136 and 403 you wrote 1100 nm. Otherwise, cite references in which the emission spectrum of the employed lamp reached that wavelength.

REPLY: Converted to 1100nm on line 18 and 68, as that is more appropriate.

Line 53. Write Gram with first uppercase letter.

REPLY: DONE on LINE 54.

Figure S1. The picture of the beers in cups should be more professional, with an uniform illumination and background.

REPLY: The pictures are only representatives, and not in the main text, so, we have retained them.

Tables S1 and S2. What is the symbol @?

REPLY: WE ARE UNSURE WHAT THIS @ REFERS TO. WE DO NOT SEE ANY SUCH SYMBOL IN TABLES S1 and S2.

Line 133. Correct fittewith.

REPLY: CORRECTED ON LINE 136.

Figure 1. Is it possible to use the same scale for figs. 1b and 1c.

REPLY: COMBINED FIGS 1b and 1c UNDER A COMMON X-AXIS

Line 169. Standard deviation of the latter fluence is missed.

REPLY: ADDED on LINE 172.

Line 179 and elsewhere. Controlled samples or control samples?

REPLY: REPLACED AT ALL INSTANCES THROUGHOUT THE MANUSCRIPT

Table 1. The equation should not be place in the table, it may be relocated in the text if needed.

REPLY: EQUATION IS RUDIMENTARY AND NOT NEEDED IN THE MAIN TEXT. IN THE TABLE 1, WE HAVE CONVERTED IT INTO A FOOTNOTE.

Line 203. “tween” or “Tween”? Which kind of Tween? Source?

REPLY: SORRY, THAT WAS AN INCOMPLETE SENTENCE, AND THE SENTENCE HAS NOW BEEN REPLACED: "An aliquot (1 mL) of the incubated unacclimatized L. brevis MRS broth was pipetted into 100 mL sterilized 85:15 medium that contains 85% beer and 15% MRS broth. "

Section 2.3. should appear before section 2.2. since bacteria were first cultured and then treated.

REPLY: WE FEEL IT IS OKAY TO LIST SECTION 2.3 LATER AS IT ONLY RELATES TO MICROBIOLOGICAL EXPERIMENTS, WHICH ARE A SUBSECTION OF THE ANALYSIS, WHILE SECTION 2.2 IS MORE ELEMENTARY AND IS RELEVANT FOR ALL SUBSECTIONS.

Line 230. Frequency of water change should be omitted.

DELETED

Lines 231-234. Calibration and cleaning details should be omitted as well as the comment on stable reading.

DELETED

Line 246. Calibration detail should be deleted.

DELETED

Lines 269-273 Details such as “Before reading, the glass on the refractometer was rinsed with distilled water and wiped with Kimwipes”, “2-3 drops of the sample were transferred on the refractometer glass using a dropper”, “the plastic cover was closed, and °Brix was read through the eyepiece” should be deleted since they are widely known.

DELETED

Line 280. Vertexing?

CONVERTED TO VORTEXING

Figure 2. Negative Y-axis values must be deleted. It seems more logic to show data ordered by increasing time.

DONE

Line 413 and elsewhere. Use italics for scientific names.

DONE

Lines 458-460. This sentence can be omitted since it follows the same logic than the previous one.

DONE on LINE 517-519: Hence, melanoidins are bigger in size and could shadow riboflavin and iso-α-acids from the UV incidence, preventing the lightstruck reaction

Reviewer 2 Report

The work entitled: “Effect of UV filters during the application of pulsed light to re- 2 duce Lactobacillus brevis contamination and 3-methylbut-2- 3 ene-1-thiol formation while preserving physico-chemical attrib- 4 utes of blonde ale and centennial red ale beers”. The authors have submitted a manuscript in which they evaluate the effect of a novel processing for preserving the quality attributes of two different beers. The article is nicely written, and well defined. Otherwise, I consider that the manuscript would need some modifications/revisions, in order to improve its overall quality.

Remarks

1. The abstract is longer than it is expected. I kindly suggest to adapt the length of this part considering the journal guidelines.

2. I would recommend to add the novelty of concept in the abstract section.

3. Considering the title/objective of this work, I suggest including some reference of UV filters in the keywords.

4.  Supplementary material: I would recommend to use the same metric system for all units.  

5. Have these filters been used previously in other applications of PL for beverages stabilization? I suggest including this information if it is available.

6. Material and Methods: (1) Was the overall experiment performed three independent times (considering three different batches of initial product)? (2) Please, indicate if all analyses were performed immediately after production or not (3) Is it possible to include more information about filters specifications (f.e. commercial supplier, commercial reference, etc.)? (4) Please, add more specific information about the statistical program (version, etc.).  (5) Table 5: I would suggest to insert the formula as table footnote. (6) Please, complete the information of Figure 1, as mentioned in the point 8. (7) If it is possible, please indicate the temperature of samples in the physicochemical analyses. 

7. Results and Discussion: Was the temperature of samples been controlled during the treatment (before/after the treatment)? Was any increase in temperatura detected? (2) Figure 2. Are bars corresponding to “c-value” correctly indicated in the Figure? Please, check it. (3) Table 2: Meaning of the acronyms 3-MBT should be indicated. (4) Table 3: Style of the footnote should be adjusted according to the style of other Tables; Are these letters (a, b, etc.) indicating significant differences between treatments in each beer style? (5) Line391-394: Please, correct the expression “p<0.05”, according to the style of this journal. (6) In some section (introduction/discussion), it would be interesting to provide specific information about the effect of conventional treatments on the parameters studied in the current study, in order to emphasize the benefits of this novel processing.

 8. General remarks. I would recommend to the authors to improve the presentation of all Figures/Tables. (1) In some figures, it is not possible to identify each image with the corresponding reference (f.e. A, B, etc.), which is mentioned in the footnote. Please, improve the identification of each content with their corresponding explanation. (2) Check if each reference/Acronym that appeared in the Images/Tables is correctly defined in the corresponding footnote. In some Figures/Tables, this information is not included.

Author Response

The work entitled: “Effect of UV filters during the application of pulsed light to re- 2 duce Lactobacillus brevis contamination and 3-methylbut-2- 3 ene-1-thiol formation while preserving physico-chemical attrib- 4 utes of blonde ale and centennial red ale beers”. The authors have submitted a manuscript in which they evaluate the effect of a novel processing for preserving the quality attributes of two different beers. The article is nicely written, and well defined. Otherwise, I consider that the manuscript would need some modifications/revisions, in order to improve its overall quality.

REPLY: WE THANK THE REVIEWER FOR THEIR TIME AND APPRECIATION OF OUR WORK>

Remarks

  1. The abstract is longer than it is expected. I kindly suggest to adapt the length of this part considering the journal guidelines.

REPLY: ABSTRACT HAS BEEN REDUCED TO 194 WORDS NOW.

  1. I would recommend to add the novelty of concept in the abstract section.

REPLY: THE NOVELTY OF THE CONCEPT HAS BEEN ADDED. "This study is the first to investigate the effect of different portions of the PL spectrum on UV-sensitive beers (light-colored Blonde Ale and dark-colored Centennial Red Ale) , using clear and bronze-tinted UV-filters. "

  1. Considering the title/objective of this work, I suggest including some reference of UV filters in the keywords.

ADDED

  1. Supplementary material: I would recommend to use the same metric system for all units.  

CONVERTED SI units to METRIC SYSTEM

  1. Have these filters been used previously in other applications of PL for beverages stabilization? I suggest including this information if it is available.

NO, THEY HAVE NOT BEEN USED FOR PL AND THIS HAS BEEN CLARIFIED IN THE ABSTRACT

  1. Material and Methods:(1) Was the overall experiment performed three independent times (considering three different batches of initial product)? (2) Please, indicate if all analyses were performed immediately after production or not (3) Is it possible to include more information about filters specifications (f.e. commercial supplier, commercial reference, etc.)? (4) Please, add more specific information about the statistical program (version, etc.).  (5) Table 5: I would suggest to insert the formula as table footnote. (6) Please, complete the information of Figure 1, as mentioned in the point 8. (7) If it is possible, please indicate the temperature of samples in the physicochemical analyses. 

REPLY: 1) YES< EXPERIMENTS WERE PERFORMED THREE INDEPENDENT TIMES WITH DIFFERENT BATCHES.

2) ALL ANALYSIS EXCEPT 3-MBT WERE PERFORMED IMMEDIATELY AFTER PRODUCTION

3) WE HAVE ADDED FURTHER INFORMATION

4) WE HAVE ADDED MORE INFORMATION

5) EQUATION HAS BEEN CONVERTED TO FOOTNOT

6) FIGURE 1 NOW HAS THE a AND b tags.

7) TEMPERATURES ARE INDICATED.

  1. Results and Discussion: Was the temperature of samples been controlled during the treatment (before/after the treatment)? Was any increase in temperatura detected? (2) Figure 2. Are bars corresponding to “c-value” correctly indicated in the Figure? Please, check it. (3) Table 2: Meaning of the acronyms 3-MBT should be indicated. (4) Table 3: Style of the footnote should be adjusted according to the style of other Tables; Are these letters (a, b, etc.) indicating significant differences between treatments in each beer style? (5) Line391-394: Please, correct the expression “p<0.05”, according to the style of this journal. (6) In some section (introduction/discussion), it would be interesting to provide specific information about the effect of conventional treatments on the parameters studied in the current study, in order to emphasize the benefits of this novel processing.

1) TEMPERATURES WERE MEASURED. MAXIMUM TEMPERATURE INCREASE WAS LESS THAN 5oC.

2) WE HAVE CORRECTED THE BARS

3) INDICATED 3-MBT FULL FORM in TABLE 2.

4) ADJUSTED TABLE 3'S STYLE AS MENTIONED.

5) WE DO NOT UNDERSTAND THE COMMENT. PLEASE CLARIFY. p < 0.05 is the statistical test that was conducted. We checked the style of the journal, but could not find any indication about what this comment meant.

  1. General remarks. I would recommend to the authors to improve the presentation of all Figures/Tables. (1) In some figures, it is not possible to identify each image with the corresponding reference (f.e. A, B, etc.), which is mentioned in the footnote. Please, improve the identification of each content with their corresponding explanation. (2) Check if each reference/Acronym that appeared in the Images/Tables is correctly defined in the corresponding footnote. In some Figures/Tables, this information is not included.

1) INCLUDED THE REFERENCE IN FIG 1.

2) CHECKED ALL ACRONYMS AND MADE SURE THEY ARE INCLUDED.

Round 2

Reviewer 1 Report

Dear authors

No new remarks from my side. Good job!

Author Response

We thank the reviewer for their time and energy invested into the manuscript.

Reviewer 2 Report

I deeply appreciate the revision conducted by the authors. 

Information about the statistical program (version, etc.) is not found in the revised version, although the authors indicate that it has been included. Please, revise this point.

Considering the authors response, I think I would be recommendable to indicate the maximum temperature that samples reached during the treatments, f.e., in M&M section.

I think that in Figure 1, one “results bar” is not displayed correctly (which corresponds to NF-5 min. Blonde Ale). Review this aspect in the final version please.  

Concerning the comment ”(5) Line391-394: Please, correct the expression “p<0.05”, according to the style of this journal”, I apologize for the confusion. I referred to correct the expression according to the style of this journal (italics, spaces, etc.).

Regarding “Point 8”, it is important that any acronym included in a Table/Figure is correctly indicated in their corresponding footer, so that  each Table/Figure can be interpreted independently of the information included in the manuscript.

Author Response

I deeply appreciate the revision conducted by the authors. 

Information about the statistical program (version, etc.) is not found in the revised version, although the authors indicate that it has been included. Please, revise this point.

RESPONSE: WE HAVE ADDED THIS INFORMATION ON LINES 303-308: "

One-way analysis of variance (ANOVA) and least significant difference (LSD) were used at α = 0.05 for microbial, color, bitterness, pH, and TSS analyses, while Two-way ANOVA and LSD were used at α = 0.05 to compare concentrations of 3-MBT using Minitab (Version 21.3.0, Minitab LLC, State College, PA, USA). A linear correlation analysis was conducted on 3-MBT calibration curves using Microsoft Excel (Microsoft Excel for Mac Version 16.69, Microsoft Corporation, Redmon, WA, USA)."

Considering the authors response, I think I would be recommendable to indicate the maximum temperature that samples reached during the treatments, f.e., in M&M section.

RESPONSE: Lines 177-182 in M&D include the require information: "Temperature was monitored using a T-type flexible wire thermo-couple (wire diameter 0.0762 mm, Omega Engineering Corp.,Stamford, Conn., U.S.A.) with the readings recorded at 1 s intervals using a data acquisition system (HP34970A, Hewlett, Packard, Loveland, Colo., U.S.A.). Temperature rise was less than 5oC for all treatments, with maximum temperature rise of 4.8 oC reported for 16.75 J/cm2 treatment without filters. "

I think that in Figure 1, one “results bar” is not displayed correctly (which corresponds to NF-5 min. Blonde Ale). Review this aspect in the final version please.  

RESPONSE: Thanks for catching the unintentional error. The data was present in the excel, but for some reason the last bar became invisible. We have fixed the error.

Concerning the comment ”(5) Line391-394: Please, correct the expression “p<0.05”, according to the style of this journal”, I apologize for the confusion. I referred to correct the expression according to the style of this journal (italics, spaces, etc.).

RESPONSE: All p < 0.05 have proper spaces and are italicized.

Regarding “Point 8”, it is important that any acronym included in a Table/Figure is correctly indicated in their corresponding footer, so that  each Table/Figure can be interpreted independently of the information included in the manuscript.

RESPONSE: We have checked and the only acronym missing we found was UV and SD, which was added. Please let us know if there is any other thing.